



# Pollution trace gases $C_2H_6$, $C_2H_2$, HCOOH, and PAN in the North Atlantic UTLS: observations and simulations

Gerald Wetzel[1], Felix Friedl-Vallon[1], Norbert Glatthor[1], Jens-Uwe Grooß[2], Thomas Gulde[1], Michael Höpfner[1], Sören Johansson[1], Farahnaz Khosrawi[1], Oliver Kirner[3], Anne Kleinert[1], Erik Kretschmer[1], Guido Maucher[1], Hans Nordmeyer[1], Hermann Oelhaf[1], Johannes Orphal[1], Christof Piesch[1], Björn-Martin Sinnhuber[1], Jörn Ungermann[2], and Bärbel Vogel[2]

[1]Karlsruhe Institute of Technology, Institute of Meteorology and Climate Research, Karlsruhe, Germany

[2]Research centre Jülich, Institute of Energy and Climate Research – Stratosphere (IEK-7), Jülich, Germany

[3]Karlsruhe Institute of Technology, Steinbuch Centre for Computing, Karlsruhe, Germany

*Correspondence to:* Gerald Wetzel (gerald.wetzel@kit.edu)

## Abstract

Measurements of the pollution trace gases ethane ($C_2H_6$), ethyne ($C_2H_2$), formic acid (HCOOH), and peroxyacetyl nitrate (PAN) were performed in the North Atlantic upper troposphere and lowermost stratosphere (UTLS) region with the airborne limb imager GLORIA (Gimballed Limb Observer for Radiance Imaging of the Atmosphere) with high spatial resolution down to cloud top. Observations were made during flights with the German research aircraft HALO (High Altitude and LOng Range Research Aircraft) in the frame of the WISE (Wave-driven ISentropic Exchange) campaign, which was carried out in autumn 2017 from Shannon (Ireland) and Oberpfaffenhofen (Germany). Enhanced volume mixing ratios (VMR) of up to 2.2 ppbv $C_2H_6$, 0.2 ppbv $C_2H_2$, 0.9 ppbv HCOOH, and 0.4 ppbv PAN were detected during the flight on 13 September 2017 in the upper troposphere and around the tropopause above the British Isles. Since PAN has the longest lifetime of this foursome, elevated quantities of this molecule could be measured even in the lowermost stratosphere (locally up to 14 km). Backward trajectory calculations as well as global three-dimensional CLaMS simulations with artificial tracers of air mass origin have shown that the main sources of the observed pollutant species are forest fires in North America and anthropogenic pollution in South and Southeast Asia uplifted and moved within the Asian monsoon anticyclone (AMA) circulation system. After release from the AMA, these species or their precursor substances are transported by strong tropospheric winds over large distances, depending on their particular atmospheric lifetime of up to months. Observations are compared to simulations with the atmospheric



models EMAC (ECHAM5/MESSy Atmospheric Chemistry) and CAMS (Copernicus Atmosphere Monitoring Service). These models are qualitatively able to reproduce the

measured VMR enhancements but underestimate the absolute amount of the increase. Increasing the emissions in EMAC by a factor of 2 reduces the disagreement between simulated and measured results and illustrates the importance of the quality of emission databases used in chemical models.

**1    Introduction**

Organic compounds in the troposphere like ethane, ethyne, formic acid, and secondary order (not directly emitted) pollutants like peroxyacetyl nitrate are involved in many atmospheric processes. These pollutants can be transported into remote regions due to their long lifetime under appropriate atmospheric conditions, like convective processes combined with strong

wind regimes of the upper troposphere such as the subtropical jet stream (e.g., Lu et al., 2019; Alvarado et al., 2020). At these altitudes, such hydrocarbons and nitrogen-containing substances (like peroxyacetyl nitrate) may also influence the amount of ozone that is known to be an effective greenhouse gas in the upper troposphere and lowermost stratosphere (UTLS) region since it largely influences the radiation budget around the tropopause (Forster and Shine,

1997; Hansen et al., 1997; Xie et al., 2008; Riese et al., 2012).

Emissions of ethane ($C_2H_6$), the most important tropospheric non-methane hydrocarbon (NMHC) species, are connected with biomass burning and natural gas losses (Rudolph, 1995; Singh et al., 2001). The production of fossil fuels together with biofuel use also are important sources of this molecule (Xiao et al., 2008). It is removed from the atmosphere by reaction with

the hydroxyl (OH) radical (Xiao et al., 2008). The mean lifetime of $C_2H_6$ is about two months (Rudolph, 1995) such that $C_2H_6$ may be transported far away from its source regions.

The trace gas ethyne ($C_2H_2$) is emitted into the troposphere by combustion of biofuels and fossil fuels, as well as biomass burning (Xiao et al., 2007). As in the case of $C_2H_6$, the reaction with the OH radical is also responsible for the loss of $C_2H_2$ in the atmosphere. The mean lifetime of

$C_2H_2$ is shorter compared to the one of $C_2H_6$ and amounts to about two weeks (Xiao et al., 2007).

Formic acid (HCOOH) has many different sources. Direct emissions from plants, biomass burning, and fossil fuel combustion are important factors producing this molecule (Mungall et


al., 2018). A secondary photochemical formation takes place from anthropogenic and biogenic

precursors (Yuan et al., 2015) like oxidation of volatile organic compounds (Khare et al., 1999). The loss of HCOOH is possible due to wet and dry deposition as well as oxidation with the OH radical (Paulot et al., 2011). The atmospheric mean lifetime of HCOOH is very variable and ranges from one or two days in the boundary layer up to a few weeks in the free troposphere (Millet et al., 2015).

The molecule peroxyacetyl nitrate ($CH_3COO_2NO_2$), commonly named as PAN, is formed via a three-body-reaction of peroxyacetyl ($CH_3COO_2$) with nitrogen dioxide ($NO_2$) and a third partner M (mainly $N_2$ or $O_2$):

$$CH_3COO_2 + NO_2 + M \leftrightarrow CH_3COO_2NO_2 + M. \tag{R1}$$

The reverse reaction of (R1) is the thermal decomposition of PAN and defines the main loss of

this molecule in the atmosphere (Fischer et al., 2014) while photolysis becomes dominant in the UTLS region (Fadnavis et al., 2015). Two minor loss processes of PAN are reactions with OH and dry deposition (Fischer et al., 2014). The mean atmospheric lifetime of PAN is very variable since it is strongly dependent on the ambient temperature. While the mean lifetime amounts to only one hour at temperatures of 298 K, it rises up to a few months under cold upper

tropospheric conditions (Singh, 1987; Fischer et al., 2014). Hence, enhanced PAN amounts (which serve as an $NO_2$ reservoir) can be transported over wide geographical regions in the middle and upper troposphere. Thus, $NO_2$ released by the backward reaction of (R1) may contribute to an increase of tropospheric ozone far away from the PAN sources (Singh, 1987; Fadnavis et al., 2014; Ungermann et al., 2016).

Scientific flights of the airborne limb imager GLORIA (Gimballed Limb Observer for Radiance Imaging of the Atmosphere) were carried out aboard HALO (High Altitude and LOng Range Research Aircraft) during the WISE (Wave-driven ISentropic Exchange) campaign in autumn 2017 above the North Atlantic. This atmospheric region is characterized by intense dynamical activity reflected in complex structures like tropopause folds and air masses of different origin.

Tropospheric pollutants like $C_2H_6$, $C_2H_2$, HCOOH, and PAN were observed together with stratospheric trace gases like $O_3$ in the UTLS region. A description of the GLORIA instrument, data analysis and chemical modelling is given in Sect. 2. A discussion of observed vertical volume mixing ratio (VMR) profiles of trace species follows in Sect. 3 together with a comparison of the measured data to simulations of the chemistry climate model EMAC

(ECHAM5/MESSy Atmospheric Chemistry) and to assimilated data of CAMS (Copernicus



Atmosphere Monitoring Service). A discussion on the possible origin of air masses at the Earth's boundary layer detected by GLORIA is also included in this section.

## 2    GLORIA instrument, data analysis and modelling

In the following subsections, we give an overview of the GLORIA instrument and the flights with the HALO aircraft, together with the corresponding data analysis and a description of atmospheric modelling performed for this study.

### 2.1    GLORIA instrument and HALO flights

The cryogenic Fourier Transform limb emission spectrometer GLORIA operates in the thermal
infrared spectral region between about 7 and 13 µm using a 2-dimensional detector array using 128 vertical and 48 horizontal interferograms per measurement (Friedl-Vallon et al., 2014; Riese et al., 2014). The interferograms are Fourier transformed into the spectral domain and radiometrically calibrated using in-flight measurements of two blackbodies (Kleinert et al., 2014). Furthermore, spectra in the horizontal direction were averaged to improve the signal to
noise ratio such that the noise equivalent spectral radiance (NESR) finally amounts to about $1.5 \times 10^{-8}$ W(cm$^2$ sr cm$^{-1}$)$^{-1}$. Spectra recorded with maximum optical path difference of 8.0 cm, which corresponds to an unapodized spectral resolution of 0.0625 cm$^{-1}$, were used for this study. These so-called chemistry mode spectra, are apodized with the Norton and Beer (1976) "strong" function. Due to the high spectral resolution, these measurements allow the retrieval of many
species with minor contribution to the spectra by the separation of individual spectral lines from continuum-like emissions.

In this study, we report results from the WISE aircraft campaign, which was dedicated mainly to the investigation of dynamical processes and the evolution of air masses within the context of stratosphere-troposphere exchange. Sixteen flights with HALO were performed from
Shannon (Ireland) and Oberpfaffenhofen (Germany) between 31 August 2017 and 21 October 2017, in a region within about 40° W and 15° E longitude and 40° N and 75° N latitude. In the following sections, we focus on results of the flight on 13 September 2017 where strong enhancements in the VMR of the pollutant species $C_2H_6$, $C_2H_2$, HCOOH, and PAN were observed by GLORIA. Figure 1 shows the flight path consisting of two main legs together with
tangent points of GLORIA where the instrument was operated in the chemistry mode.



## 2.2 Data analysis of measured spectra

Radiances are calculated with the Karlsruhe Optimized and Precise Radiative transfer Algorithm (KOPRA; Stiller et al., 2002) based on spectroscopic parameters from the high-resolution transmission molecular absorption database (HITRAN; Gordon et al., 2017). The retrieval itself is performed with the procedure KOPRAFIT (Höpfner et al., 2002) using derivatives of the radiance spectrum with respect to atmospheric state and instrument parameters (Jacobians) calculated by KOPRA. The inverse problem of radiative transport is solved by KOPRAFIT with a Gauss-Newton iterative method (Rodgers, 2000) in combination with a Tikhonov-Phillips regularization approach (Phillips, 1962; Tikhonov, 1963) using a constraint with respect to a first derivative of the a priori profile of the target species.

In a first step cloud affected spectra are filtered out using a cloud index as described in Spang et al. (2004). Spectra with a colour ratio of the mean radiance in two spectral windows (788.20-796.25 $cm^{-1}$ and 832.30-834.40 $cm^{-1}$) larger than 2 were chosen for the retrieval process. Prior to the trace gas retrievals, the pointing elevation angle was retrieved to compensate for systematic misalignment of the line of sight (LOS) of the instrument according to the method described in Johansson et al. (2018). The final pointing (LOS) error was estimated to 0.01°. Vertical profiles of atmospheric species were taken from a climatological atmosphere (Remedios et al., 2007), updated with surface concentration data from NOAA ESRL GMD (National Oceanic and Atmospheric Administration, Earth System Research Laboratory, Global Monitoring Division; Montzka et al., 1999). Subsequently, a temperature retrieval was carried out using pressure-temperature a priori data from the European Centre for Medium-Range Weather Forecasts (ECMWF) which was interpolated to the GLORIA vertical retrieval grid. Spectral windows around 811 and 957 $cm^{-1}$ containing appropriate $CO_2$ transitions were used for this analysis. The total error of the temperature retrieval calculations is estimated to about 1.5 K (Johansson et al., 2018).

The spectral analysis of the target species $C_2H_6$, $C_2H_2$, HCOOH, and PAN is impeded by overlapping features of so-called disturbing gases in the corresponding spectral region. Hence, a careful selection of appropriate microwindows is essential to perform retrievals with good accuracy. Test retrievals allowed to find microwindows with respect to little overlapping of spectral signatures of disturbing gases in combination with a high sensitivity of line intensities of target species with respect to changes in their concentration. Main interfering species were



either adjusted simultaneously together with the target molecule or pre-fitted in a different spectral interval.

The analysis of $C_2H_6$ was performed in four microwindows within the $v_9$ band between 819 and

833 cm$^{-1}$ (see Fig. 2a). Many overlapping features of so-called disturbing gases are visible in this spectral region, first of all $H_2O$ and $CO_2$. The error budget of $C_2H_6$ is shown in Fig. 2b. At higher altitudes with low $C_2H_6$ values, the total error of this molecule is dominated by the temperature error. Lower down in the VMR maximum region, the systematic spectroscopic error of $C_2H_6$ (as given in HITRAN) governs the error budget. The total error of $C_2H_6$ remains

within about 15 % in the altitude region of the upper troposphere.

Retrieval calculations of $C_2H_2$ were carried out in the R-branch region of the $v_5$ band. Figure 3a shows spectral contributions of relevant species in four microwindows between 759 and 781 cm$^{-1}$ that have been found best appropriate to derive $C_2H_2$ from GLORIA spectra. The corresponding retrieval error budget of $C_2H_2$ is given in Fig. 3b. The random noise error is

dominating the budget over nearly the complete altitude range. The total error of $C_2H_2$ stays within 10-15 % in the region of the VMR maximum in the upper troposphere.

The analysis of the molecule HCOOH was performed in the spectral range between 1086 and 1117 cm$^{-1}$ (see Fig. 4a). Three microwindows were chosen including the strong Q-branch of the HCOOH $v_6$ vibrational band. The spectral region is dominated by spectral features due to

$O_3$, $CO_2$, CFC-12, and HCFC-22 transitions. The spectroscopic part of the total error is dominant in the altitude region of the HCOOH VMR maximum in the upper troposphere (see Fig. 4b). Here, the total HCOOH error stays within 10 %. Apart from this altitude region, the error budget is governed by the random noise part and the total error increases significantly (mainly in the upper part of the profile with low HCOOH values).

The retrieval of PAN was conducted in two broad microwindows of the $v_{16}$ band between 780 and 805 cm$^{-1}$ (see Fig. 5a). The spectral gap between these microwindows was chosen such that the strong $CO_2$ Q-branch at 792 cm$^{-1}$ was excluded from the data analysis. Beside $CO_2$, the molecules $H_2O$ and $CCl_4$ are the main interfering species in the spectral region of PAN. The retrieval error budget is depicted in Fig. 5b. Beside random noise, further error sources like

inaccuracies in the FOV and the retrieved temperature profile contribute to the total PAN error that remains within 10 % in altitude regions with enhanced PAN amounts.

Besides the retrieval of the above-mentioned pollutant gases, the tracer species ozone was also inferred from the recorded spectra. Many spectral ozone lines are available in the mid-infrared



spectral region. Transitions between 780 and 788 cm$^{-1}$ within the $\nu_2$ band were chosen for the

retrieval process similar to the method described in Johansson et al. (2018). The total ozone error is within 10 % with a vertical resolution of 0.3 to 1.5 km.

The altitude resolution of all retrievals, calculated from the full width at half maximum of the columns of the averaging kernel matrix, was used as an a posteriori quality filtering of the retrieved data. Only vertical profile parts with an altitude resolution of better than 2 km were

finally used for the data interpretation.

### 2.3    Model simulations

#### 2.3.1    EMAC

Retrieved vertical profiles of trace species are compared to a multi-year simulation of the chemistry climate model ECHAM5/MESSy Atmospheric Chemistry (EMAC). This Eulerian

model includes submodels describing tropospheric and middle atmosphere processes (Jöckel et al., 2010). The core model is the 5th generation European Centre Hamburg general circulation model (ECHAM5; Roeckner et al., 2006) that is connected to the submodels using the interface Modular Earth Submodel System (MESSy). For the present study we applied EMAC (ECHAM5 version 5.3.02, MESSy version 2.53) with a spherical truncation of T106

(corresponding to a resolution of approximately 1.125 by 1.125 degrees in latitude and longitude) with 90 hybrid pressure levels from the ground up to 0.01 hPa. Meteorological data fields are specified using a Newtonian relaxation technique of the surface pressure and prognostic variables below 1 hPa with the ECMWF reanalysis ERA-Interim (Dee et al., 2011). The simulation was initialized on 1 May 2017 and includes a comprehensive chemistry setup

from the troposphere to the lower mesosphere. Rate constants of gas-phase reactions originate from Atkinson et al. (2007) and Sander et al. (2011). Photochemical reactions of precursor substances important for the build-up of the species PAN (Fischer et al., 2014) were integrated into the model setup. For surface emissions of non-methane volatile organic compounds (NMVOCs), a data set of the MACCity emission inventory (MACC/CityZEN; Granier et al.,

2011) and ACCMIP (Atmospheric Chemistry and Climate Model Intercomparison Project; Lamarque et al., 2013) was used. Anthropogenic emission sources from biomass burning, agricultural waste burning, fossil fuels, ship, road and aircraft, as well as biogenic emissions are considered. For the simulated year 2017, most recent available emissions of 2010 are repeated. In addition to this EMAC standard run, a second model simulation (called EMAC_2)



was performed using NMVOC emissions enhanced by a factor of 2 as recommended by Monks
       et al. (2018). The model output data were saved every 5 hours during the time period of the
       GLORIA observations. The model output to the GLORIA measurements was interpolated in
       time and space to the observation geolocations.

### 2.3.2   CAMS

The Copernicus Atmosphere Monitoring Service (CAMS) produced by ECMWF is a reanalysis
       dataset that produces continuous data on atmospheric composition (Inness et al., 2019). The
       Integrated Forecast System (IFS) of ECMWF was integrated to allow for the data assimilation
       and modelling of aerosols, chemically reactive species and greenhouse gases. Apart from
       assimilated ozone, no stratospheric chemistry is simulated by the model system. In this study,
CAMS reanalyses were used with a horizontal resolution of about 80 km. The vertical
       resolution consists of 60 pressure levels up to 0.1 hPa. Three-dimensional model output fields
       are available every 3 hours. Detailed information on the CAMS model architecture is given by
       Inness et al. (2019). An evaluation study of CAMS using aircraft observations was carried out
       by Wang et al. (2020). Biases of assimilated species like ozone are found to be less than 20%
whereas discrepancies for gases like $C_2H_6$ and PAN are generally larger.

## 3    Results and discussion

In this section, vertical profiles retrieved from GLORIA measurements during the WISE
campaign on 13 September 2017 over the North Atlantic region are shown. Observed GLORIA
chemistry mode data are compared to EMAC and CAMS simulation results. The possible origin
of air masses detected by GLORIA is also discussed.

### 3.1    GLORIA measurements

Retrieved volume mixing ratios of $C_2H_6$, $C_2H_2$, HCOOH, and PAN together with $O_3$ inferred
from limb emission spectra during the WISE flight on 13 September 2017 are displayed in Fig.
6. Ozone is a molecule with highest concentrations in the stratosphere (Brasseur and Solomon,
2005). Hence, it can be used as a tracer to diagnose detected air masses whether they are of
stratospheric or tropospheric origin. The general shape of $O_3$ VMR is strongly correlated with
the tropopause as shown in Fig. 6a. There are two regions where stratospheric air comes down
to about 7 km (around 14:45 UTC and around 16:50 UTC). Here, the tropopause layer reaches



these low altitudes in the form of a stratospheric intrusion, while at the beginning and the end of the measurement phase the troposphere extends up to about 12 km. Trajectory calculations have shown that due to a west-southwesterly mid- and upper tropospheric air flow in the region of the flight path shown in Fig. 1, GLORIA has sounded virtually the same air mass twice yielding to a kind of symmetry in the horizontal trace gas distribution before and after 16:10

UTC. Measured stratospheric ozone volume mixing ratios are within 0.1 and 0.8 ppmv. These values are in line with other mid-latitude observations performed in this altitude range (e.g., Cortesi et al., 2007; Livesey et al., 2008).

Measured concentrations of the species $C_2H_6$ are shown in Fig. 6b. The VMR distribution of $C_2H_6$ is in parts anti-correlated to the one of ozone. Two regions of strong VMR enhancements

up to about 2.2 ppbv can be seen in the upper troposphere at the beginning and at the end of the measurement period over the region south of Ireland and near the coastline of The Netherlands and Belgium. In the stratosphere, no stronger enhanced $C_2H_6$ VMR levels are visible and values remain below about 0.6 ppbv what can be confirmed by mid-latitude satellite measurements (Rinsland et al., 2005; Glatthor et al., 2009; Wiegele et al., 2012).

The same picture is present in the vertical and horizontal distribution of $C_2H_2$ amounts (see Fig. 6c). Elevated volume mixing ratios of up to 0.2 ppbv in the upper troposphere are clearly visible at the beginning and close to the end of the observation period. In the stratosphere, measured $C_2H_2$ values appear noisy and stay clearly below 0.1 ppbv most of the time. $C_2H_2$ VMR measured by GLORIA lies within the range of satellite data obtained in the same altitude region

at mid-latitudes (Rinsland et al., 2005; Wiegele et al., 2012).

Measured HCOOH volume mixing ratios are depicted in Fig. 6d. As in the case of the previously mentioned species $C_2H_6$ and $C_2H_2$, large amounts of HCOOH are also visible at high altitudes in the troposphere during early and late time of the GLORIA observations with values up to 0.9 ppbv. In contrast, stratospheric HCOOH values are low and not higher than 0.1 ppbv.

These values are in accordance with space-borne mid-latitude observations in the altitude regime considered here (Rinsland et al., 2006; Grutter et al., 2010).

The 2-dimensional cross section of PAN is shown in Fig. 6e. The distribution of VMR maxima and minima is more structured compared to the one of the previously regarded pollution trace gases. Increased amounts of PAN up to 0.4 ppbv are not only visible at the beginning and end

of the observation period in the upper troposphere but also in the lowermost stratosphere around 15:00 UTC, at altitudes of 7 to 8 km. Somewhat less enhanced quantities are noticeable near 14





km around 15:30 UTC. Elevated PAN amounts of comparable magnitude have also been detected in the UTLS region by space-borne instruments (Coheur et al., 2007; Wiegele et al., 2012; Ungermann et al., 2016). This different shape of VMR distribution might be explained

by very long lifetimes of PAN under cold UTLS conditions and by the fact that PAN is not emitted directly but dependent on the availability of precursor substances as described in Sect. 1.

### 3.2 Comparison to model simulations

The comparison of measured species to model simulations is presented in Figs. 7 and 8. The

observed data have been temporally smoothed with a 39-point adjacent averaging routine to permit a more realistic comparison with respect to different horizontal resolutions in the measurement and the EMAC and CAMS simulations. Concerning $O_3$, both simulations principally reproduce the tropospheric and stratospheric concentrations seen by GLORIA (see Fig. 7a-c) but with coarser spatial structure. The chemistry climate model EMAC is able to

simulate finer structures while CAMS only produces a smooth distribution of assimilated ozone. Both models tend to slightly overestimate the amount of ozone in the troposphere. This is also visible in Fig. 8a-c where differences between both EMAC runs (with and without enhanced NMVOC emissions) and GLORIA observations are shown. The amount of simulated ozone in the EMAC_2 run is only slightly higher (less than 10 ppbv) compared to the EMAC

simulation without enhanced NMVOC emissions (VMR differences in Fig. 8b and Fig. 8c are therefore nearly the same).

The comparison of $C_2H_6$ is displayed in Fig. 7d-f. Both models are able to qualitatively reproduce the temporal and spatial region of enhanced upper tropospheric $C_2H_6$ as observed by GLORIA. As in the case of ozone, EMAC again is able to display finer structures in the vertical

and horizontal distribution of $C_2H_6$ compared to CAMS. However, deficits in the simulated absolute $C_2H_6$ quantities are clearly visible in both models, especially in the case of CAMS. A considerable underestimation of CAMS $C_2H_6$ with respect to airborne observations was already reported by Wang et al. (2020). The EMAC_2 simulation with increased NMVOC emissions at least reduces the difference to the GLORIA observations compared to the EMAC run without

these stronger NMVOC emissions (see Fig. 8d-f).

Concerning the molecule $C_2H_2$, we recognize that EMAC generates elevated concentrations in the same upper tropospheric region as they are observed by GLORIA (see Fig. 7 g-h). In



addition, the measured VMRs in the upper troposphere are only little underestimated in terms of their absolute amount by the EMAC_2 simulation using raised NMVOC emissions. In the

stratosphere, simulated $C_2H_2$ amounts are too low compared to the measurement. Using standard NMVOC emissions in EMAC leads to an increased underestimation of $C_2H_2$ amounts compared to GLORIA (see Fig. 8g-i).

The comparison of the species HCOOH is shown in Fig. 7i-k and Fig. 8j-l. Elevated HCOOH concentrations, as recorded by GLORIA in the upper troposphere, are clearly underestimated

by both models, especially in the CAMS simulation, although the atmospheric region of the (too weak) enhanced HCOOH amounts in the models agrees with the measured one. However, the EMAC_2 simulation at least reduces differences with respect to the GLORIA observations.

Looking at the temporal and spatial distribution of PAN the situation appears somewhat different to the comparisons discussed above (see Fig. 7l-n and Fig. 8m-o). The principal

behaviour of enhanced PAN values in the upper troposphere is captured by both atmospheric models. EMAC produces slightly finer structures in the stratosphere compared to CAMS. However, the measured small scale variations in the amount of PAN especially near 14:30 UTC between 6 and 8 km are not reproduced by the EMAC_2 simulation with enhanced NMVOC emissions while the observed elevated PAN values around 15:00 UTC in the lowermost

stratosphere from 7 to 8 km are also visible in the model output. Apart from the regions with the highest measured PAN amounts, EMAC_2 tends to overestimate the concentration of PAN below about 13 km (what is not the case in the standard EMAC run). Interesting VMR variations are also seen by GLORIA in the stratosphere above 13 km. The PAN VMR maximum detected around 15:40 UTC near 14 km is not visible in the model simulations. The PAN VMR

minima near 15:00 UTC and between 16:00 UTC and 16:15 UTC at about 13.5 km are reflected as a VMR minimum in EMAC_2, although with lower absolute quantities. The different shape of the horizontal and vertical distribution of PAN VMR is most probably caused by long-range atmospheric transport. Compared to the species discussed before, sources and sinks are different, and atmospheric lifetime of PAN is considerably longer. In the following subsection,

we will focus on the origin of the polluted air masses, which have been detected by GLORIA.

### 3.3 Origin of polluted air masses

To estimate the geographical region (within the Earth's upper planetary boundary layer) of the origin of the measured enhanced amounts of the pollutants we performed backward trajectory



calculations as well as global three-dimensional CLaMS simulations with artificial tracers of
air mass origin as described in the following subsections.

### 3.3.1   CLaMS backward trajectory calculations

To obtain a more detailed insight into the origin and transport pathways of air masses, backward
trajectories with the three-dimensional Chemical Lagragian Model of the Stratosphere
(CLaMS; McKenna et al., 2002b; McKenna et al., 2002a; Pommrich et al., 2014) were
performed starting from the GLORIA measurements. Although pure trajectories do not include
mixing processes, they are well suited to analyse the history of transport pathways of air parcels
in the tropics and in the region of the Asian monsoon into the UTLS (Vogel et al., 2014; Li et
al., 2018; Ploeger et al., 2012). In this study, 20-, 40-, and 60-days diabatic backward
trajectories with a horizontal resolution of 1 x 1 degrees were calculated using ERA-interim
reanalysis wind data (Dee et al., 2011).

These trajectories were generated for defined areas, where enhanced or low VMRs of pollutants
have been detected by GLORIA. Selected regions are displayed as coloured boxes in Fig. 6.
High amounts of pollutants are recorded within the cyan and blue boxes mainly in the upper
troposphere. Air masses marked in these two boxes are located near the stratospheric intrusion.
This region was probed twice, first at the beginning and second at the end of the flight (see Fig.
1). In addition, the green box marks enhanced quantities of observed PAN found in the
stratosphere between 13 and 14.5 km (Fig. 6e). In contrast, the black box stands for an air mass
where low pollution VMRs have been observed.

Fig. 9 gives an overview of the trajectory calculations. It is obvious that air masses were
transported by westerly winds to the place of GLORIA observations. If we first regard the black
box (as defined in Fig. 6) in the lowermost stratosphere, where low mixing ratios for all
pollutant species discussed here were observed, we find that only few trajectories penetrate the
upper planetary boundary layer (PBL) limit of 800 hPa on their way back from the GLORIA
observation points within 20-, 40-, and 60 days (Fig. 9a-c). Further, most of these areas are
located over the southern part of the North Pacific where we would not expect much pollution
in the PBL. In contrast, looking at the cyan and blues boxes (of Fig. 6) marking mainly air
masses in the upper troposphere near the flank of the stratospheric intrusion with generally high
amounts of the pollutants, we find lots of trajectories going into the PBL not only over the North
Pacific region but also over densely populated regions in Southeast Asia (Fig. 9d-i) where we
expect direct anthropogenic emissions or precursors of the considered species (Lelieveld et al.,



2001). Furthermore, marked areas of PBL penetration are also visible over the North American continent, especially in Canada where forest fires were frequent and intense in August and September 2017 (Pumphrey et al., 2020; Torres et al., 2020; Hooghiem et al., 2020; Khaykin et al., 2018). These widespread Canadian fires correlate well with the marked PBL areas. Finally,

we look at the green region in the stratosphere around 14 km (see Fig. 6) where enhanced amounts of PAN are visible in contrast to the non-elevated values of $C_2H_6$, $C_2H_2$, and HCOOH. The corresponding trajectory calculations (Fig. 9j-l) exhibit no PBL penetration areas in the case of 20- and 40-days backward trajectories. However, the 60-days backward calculations clearly show areas over densely populated Southeast Asia where trajectories entered the PBL.

The ascending air masses are clearly visible in the changing colour of the potential temperature along the trajectories (Fig. 9l). This potential source region is located well within the Asian summer monsoon pollution pump (Lelieveld et al., 2018; Randel et al., 2010). Since PAN has a lifetime of up to a few months in the free and upper troposphere which is longer than the one of $C_2H_6$, $C_2H_2$, and HCOOH, it is likely that some amounts of PAN are still existing in the

stratospheric region of the green box while concentrations of the three other pollutant species are already depleted.

Backward trajectory calculations are very useful to identify both the origin of an air parcel in the PBL and its detailed transport pathways, however mixing processes between different air parcels are neglected. Therefore, we use in addition global three-dimensional CLaMS

simulations considering mixing of air parcels to characterize the origin of air masses.

### 3.3.2    Artificial tracers of air mass origin calculations

The Lagrangian three-dimensional chemistry transport model CLaMS (Pommrich et al., 2014, and references therein) was used to calculate artificial tracers of air mass origin (Vogel et al., 2016; Vogel et al., 2019). These artificial tracers refer to marked geographical regions in the

boundary layer of the global atmosphere. An overview of these regions is given in Fig. 10 which is an updated configuration compared to previous studies using artificial tracers of air mass origin in CLaMS. The upper limit of the model boundary layer follows the orography and extends to about 2-3 km above the Earth's surface. In the currently used simulation, the model dynamics is driven by horizontal winds from the ERA-interim reanalysis (Dee et al., 2011)

provided by ECMWF. Transport of air masses from the model boundary into the free troposphere and above is considered from 1 May 2017, the starting time of the simulation. Every 24 h (time step for mixing in CLaMS), air masses in the model boundary layer are marked



by the different tracers of air mass origin and can be transported like a chemical tracer to other regions of the free troposphere or stratosphere and subsequent mixing processes between different air masses can occur. Therefore, the value of the individual artificial tracer of air mass origin counts the percentage of an air mass that originated in the specific model boundary layer region since 1 May 2017 considering advection and mixing processes.

The results of the CLaMS simulation are displayed in Fig. 11. The origin of air masses seen inside the cyan and blue boxes (which contain the largest values of $C_2H_6$, $C_2H_2$, HCOOH, and PAN, see Fig. 6) below the tropical side of the stratospheric intrusion stem to a large part from North America, the Northeast and Northwest Pacific and the Tropical Eastern Pacific; to a smaller part also from Central America. Up to about 40 % in these boxes originate from the South Asian region which includes Tibetan Plateau, Eastern China, Northern India, Indian Ocean, Bay of Bengal, Indian subcontinent, Southeast Asia, and Warm pool (see Fig. 10). This is consistent with the findings of the trajectory calculations as shown in the previous section. Air masses in the stratosphere within the green box where enhanced PAN mixing ratios were detected by GLORIA come mainly from the South Asian region. This is also in agreement with the PBL penetration region of the backward trajectories as discussed before. Concerning the black box in the stratosphere with low concentrations of $C_2H_6$, $C_2H_2$, HCOOH, and PAN, it is obvious that only fragments of air originate from the North American, Northeast and Northwest Pacific region. Some patches of South Asian air masses are visible in the black box zone, obviously from regions without enhanced amounts of the pollutants discussed here.

The artificial tracers of air mass origin mark specific geographical regions in the model boundary layer and are therefore very useful to identify the origin of observed air masses including atmospheric mixing processes. However, the real regions on the Earth's surface with high emissions of chemical tracers such as $C_2H_6$, $C_2H_2$, HCOOH, and PAN (or their precursors) are not included in the CLaMS simulations. Nevertheless, CLaMS simulations are useful to show that the enhanced PAN mixing rations in the lower stratosphere (green box) are mainly from South Asia in agreement to the trajectory calculations.

## 4    Conclusions

GLORIA trace gas observations shown in this work were performed during a flight of the WISE aircraft campaign around the British Isles on 13 September 2017. The following main results can be stated:



First, enhanced volume mixing ratios of the pollutant gases $C_2H_6$, $C_2H_2$, HCOOH, and PAN
were recorded by the GLORIA instrument in the upper troposphere with high spatial resolution.
These enhancements were detected far away from the emission sources of these species. This
is possible due to their long atmospheric lifetimes in the order of weeks to months under free
tropospheric conditions. Since PAN has the longest lifetime of this foursome, elevated

quantities of this molecule could be measured even in the lowermost stratosphere. The main
sources of the emitted species are on the one hand biomass forest fires in North America which
reached their maximum a couple of weeks before the GLORIA flight (Pumphrey et al., 2020;
Torres et al., 2020) when air masses detected by GLORIA passed this region. On the other
hand, another important source region is located in the vast region of South and Southeast Asia

where the Asian monsoon anticyclone governs the circulation regime during the summer
months. Here, huge amounts of pollutants are lifted upwards into the upper troposphere and
further transported to northern midlatitudes via strong wind fields like the subtropical jet stream
(see, e.g., Lelieveld et al., 2018; Legras and Bucci, 2020). Indeed, another GLORIA
measurement carried out about 6 weeks earlier on 31 July 2017 during the StratoClim

(Stratospheric and upper tropospheric processes for better climate predictions) campaign over
India and Nepal also shows elevated amounts of more than 0.2 ppbv of $C_2H_2$, more than 0.2
ppbv of HCOOH, and more than 0.5 ppbv of PAN (Johansson et al., 2020). These values are of
comparable magnitude to the ones observed during the WISE campaign.

Second, the chemistry climate model EMAC and the CAMS assimilation system are able to
simulate tropospheric and stratospheric dynamical ozone VMR structures as seen by GLORIA
although (primarily CAMS) with coarser spatial resolution compared to the measurement. In
addition, both models reproduce the temporal and spatial region of enhanced upper tropospheric
VMR levels of the measured pollutant species ($C_2H_2$ not available in CAMS). However, CAMS
clearly underestimates the amount of elevated $C_2H_6$, HCOOH, and to a lesser extent, PAN. The

EMAC_2 simulation using NMVOC emissions enhanced by a factor of 2 (as recommended by
Monks et al., 2018) only slightly underestimates the $C_2H_2$ concentrations while simulated
values for $C_2H_6$ and HCOOH are also too low. Compared to the EMAC standard emission run,
the EMAC_2 simulation reduces differences to the GLORIA observations for these gases. In
contrast, the size of elevated PAN values is overestimated by the EMAC_2 model run.

However, this does not hold for the local PAN VMR enhancements detected near 14 km, which
are neither captured by EMAC_2 nor by CAMS.



This study has shown that observations of pollutant species are further needed since biomass burning and wildfires will still occur in the future and seem to have increased in the last years (Witze, 2020). There is still potential to improve chemical models with regard to reproduce the
measured VMR enhancements of the pollutant gases in more detail. One aspect is to improve the emission databases in the models because the simulated amount of pollution species is strongly dependent on the local emission place and the intensity of the emissions.

*Data availability.* GLORIA measurements are available in the database HALO-DB
(https://halo-db.pa.op.dlr.de/mission/96) and will be available on the KITopen repository. The CAMS model data is available from ECMWF (https://apps.ecmwf.int/data-catalogues/cams-reanalysis). EMAC and CLaMS data are available upon request.

*Author contributions.* GW wrote the paper and performed the bulk of the data analysis, with
input from all co-authors. SJ, AK, JU, MH, and NG performed the GLORIA data processing. FFV, TG, EK, GM, HN, and CP operated GLORIA during the WISE campaign in Shannon and Oberpfaffenhofen. OK and FK performed the EMAC simulations and designed the sensitivity studies. BV and JUG performed the CLaMS trajectory- and artificial tracers of air mass origin calculations. BMS, HO, and JO led the funding application and directed the flight planning and
research. All authors commented on and improved the manuscript.

*Competing interests.* The authors declare that they have no conflict of interest.

*Special issue statement.* This article is part of the special issue "WISE: Wave-driven isentropic
exchange in the extratropical upper stratosphere and lower stratosphere (ACP/AMT/WCD inter-journal SI), 2019". It is not associated with a conference.

*Acknowledgements.* We acknowledge support by the German Research Foundation (Deutsche Forschungsgemeinschaft, DFG Priority Program SPP 1294). We are grateful to the WISE
coordination team for excellently conducting the aircraft campaign. Results are based on the efforts of all members of the GLORIA team, including the technology institutes ZEA-1 and



ZEA-2 at Forschungszentrum Jülich and the Institute for Data Processing and Electronics at the Karlsruhe Institute of Technology. We would also like to thank the pilots and ground-support team at the Flight Experiments facility of the Deutsches Zentrum für Luft- und Raumfahrt (DLR-FX). We thank ECMWF for providing CAMS data. The EMAC simulations were performed on the supercomputer ForHLR funded by the Ministry of Science, Research and the Arts Baden-Württemberg and by the Federal Ministry of Education and Research. The CLaMS activities contribute to the DFG project AMOS (HALO-SPP 1294/VO 1276/5-1) funded by the German Research Foundation (Deutsche Forschungsgemeinschaft, DFG). The authors gratefully acknowledge the computing time for the CLaMS simulations granted on the supercomputer JURECA at Jülich Supercomputing Centre (JSC) under the VSR project ID JICG11.

*Financial support.* We acknowledge support by Deutsche Forschungsgemeinschaft and Open Access Publishing Fund of Karlsruhe Institute of Technology.





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

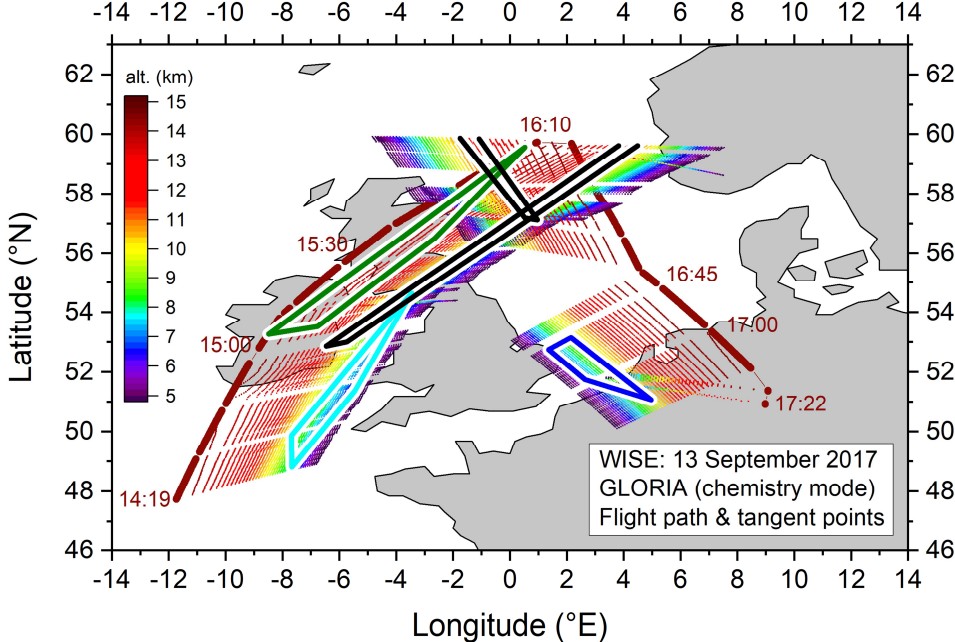

**Figure 1.** Path of the HALO flight on 13 September 2017 during the WISE campaign (large dark brown points) together with GLORIA tangent points (small points with changing colour according to altitude). Measurement times are given in UTC. Coloured framed zones mark areas of special interest as discussed in Sect. 3. Note that only observations performed in the chemistry mode are shown.

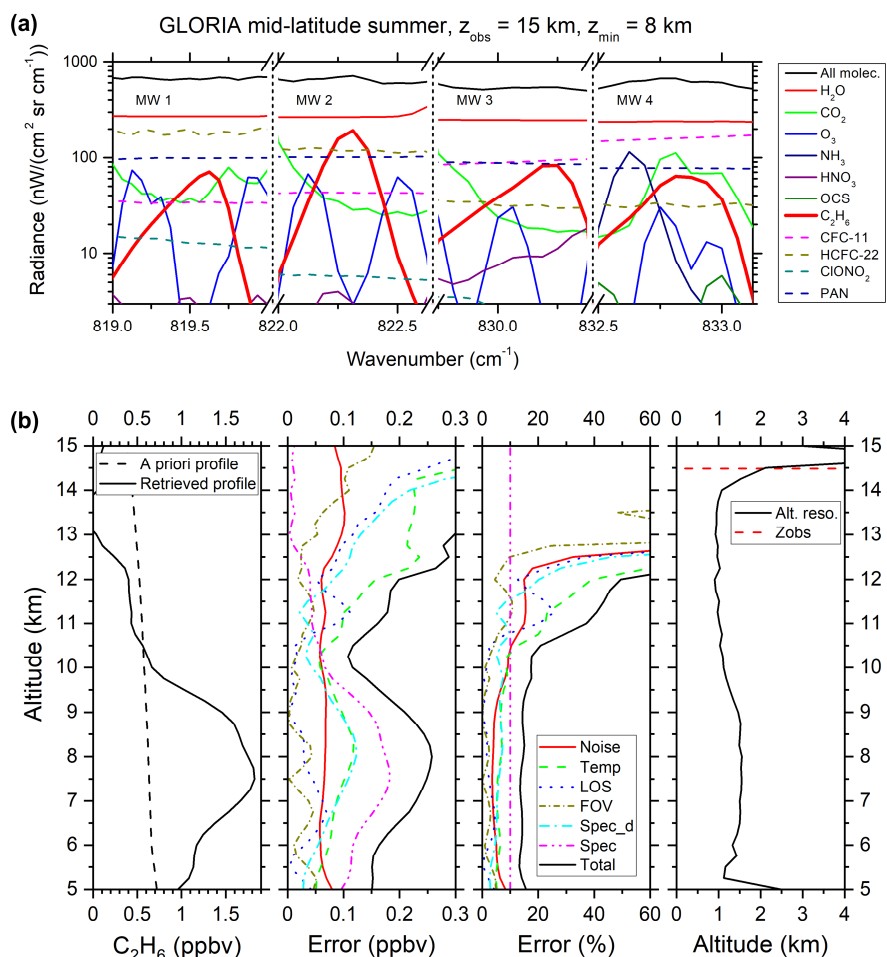

**Figure 2.** Simulated limb emission spectra (with spectral resolution of GLORIA) for a mid-latitude summer standard atmosphere (Remedios et al., 2007) in four microwindows in the spectral region of the $C_2H_6$ $\nu_9$ band centred at 822 cm$^{-1}$ for a tangent altitude of 8 km. Emissions of individual species contributing to the combined spectrum (all molecules, black line) are shown **(a)**. Retrieved $C_2H_6$ VMR vertical profile (and a priori profile) of the limb sequence measured at 13 September 2017 at 16:55 UTC combined with absolute and relative errors and the altitude resolution (alt. reso.), determined from the full width at half maximum of the columns of the averaging kernel matrix, together with the observer altitude ($z_{obs}$). The following individual $1\sigma$ errors are shown: spectral noise (red solid line), temperature (green dashed line), line of sight (LOS; blue dotted line), field of view (FOV; dark yellow short dash dotted line) spectroscopic data of disturbing gases (cyan dash dotted line), spectroscopic data of target molecule $C_2H_6$ (dash dotted magenta line), and total error (black solid line) **(b)**.


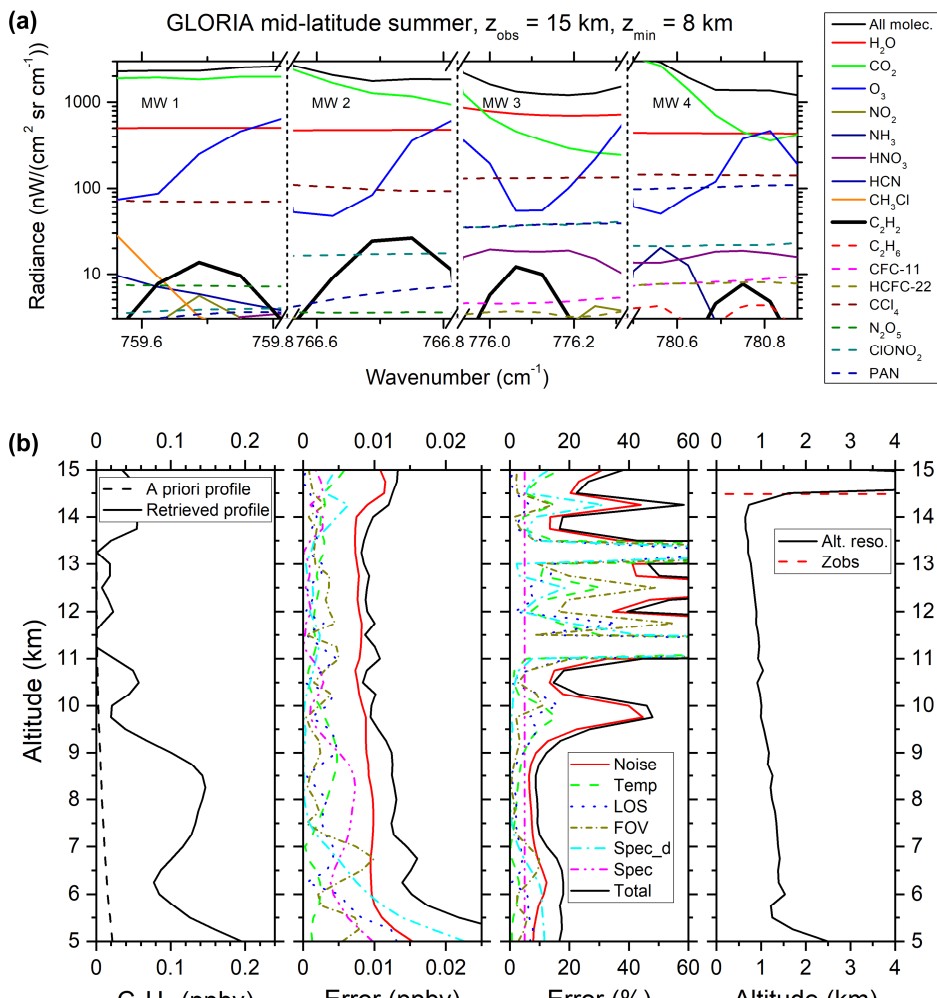

**Figure 3.** Simulated limb emission spectra for four microwindows within the $C_2H_2$ $v_5$ band centred at 730 cm$^{-1}$ for a tangent altitude of 8 km **(a)** and the error budget for a $C_2H_2$ vertical profile obtained on 13 September 2017 at 16:55 UTC **(b)**. Annotation as per Fig. 2.




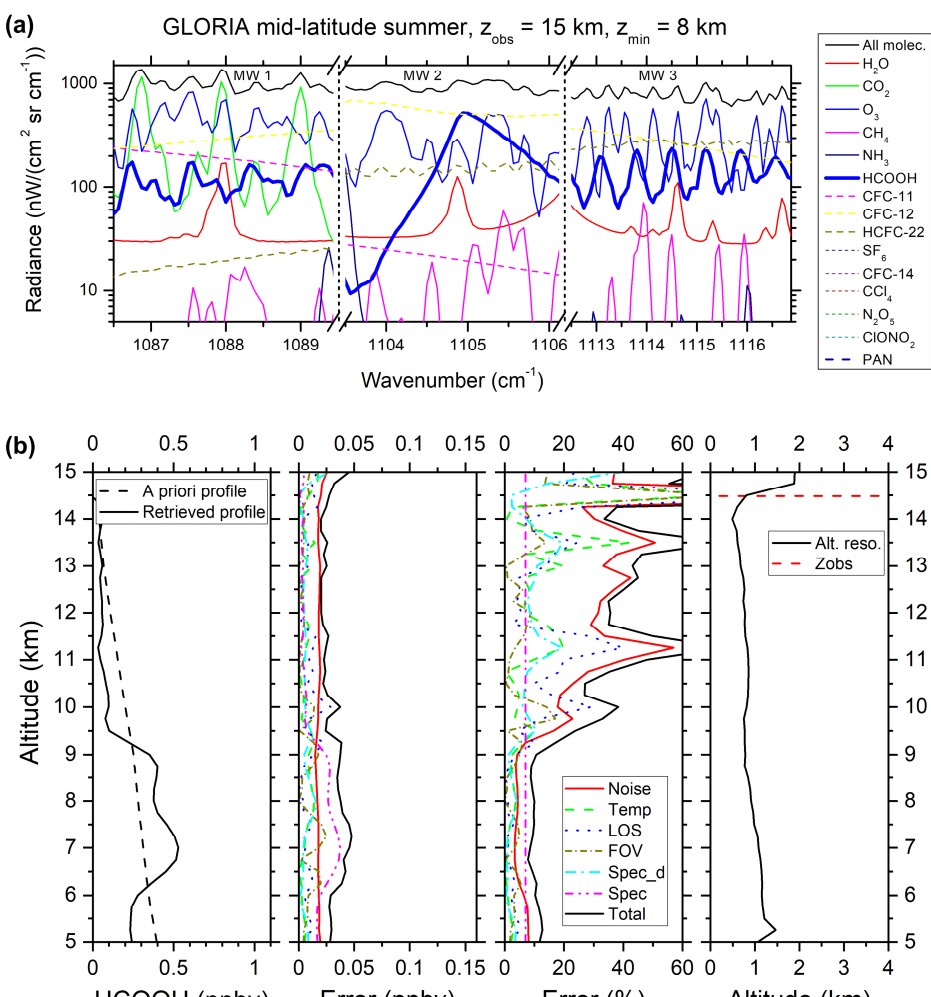

**Figure 4.** Simulated limb emission spectra for three microwindows within the HCOOH $\nu_6$ band centred near 1105 cm$^{-1}$ for a tangent altitude of 8 km **(a)** and the error budget for a HCOOH vertical profile obtained on 13 September 2017 at 16:55 UTC **(b)**. Annotation as per Fig. 2.

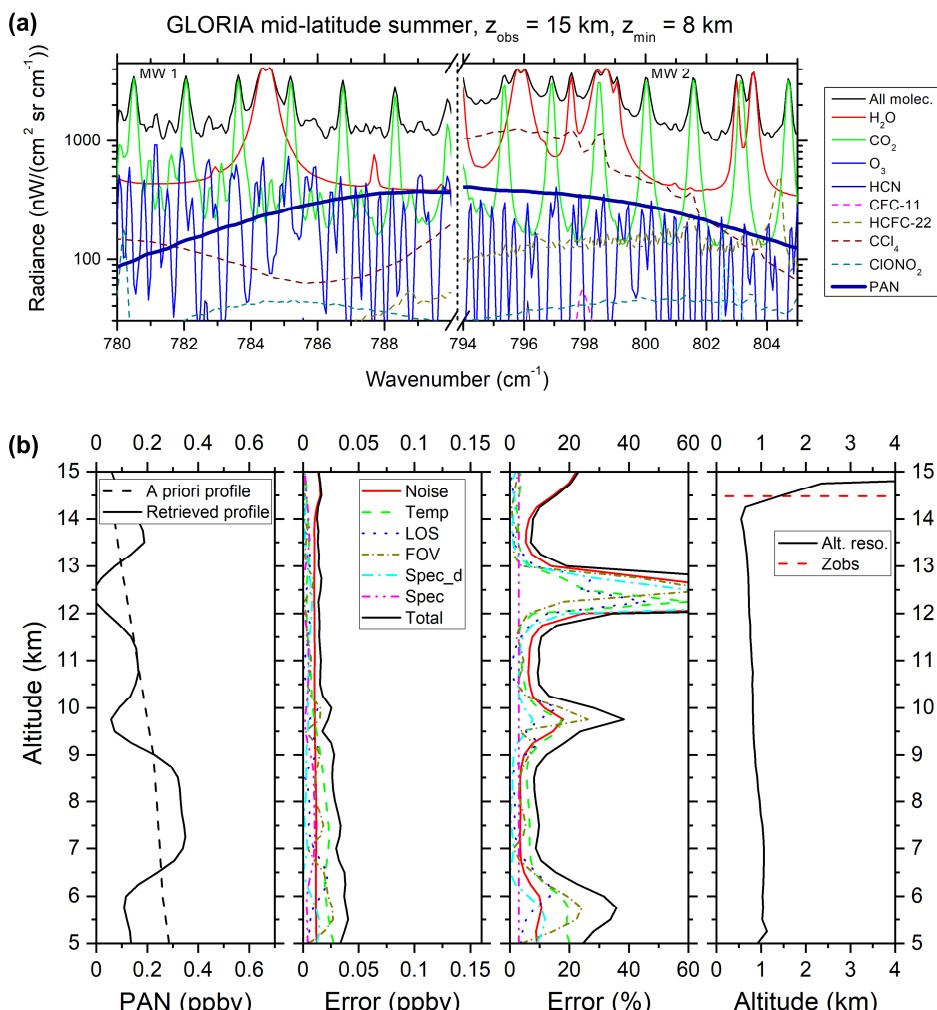

**Figure 5.** Simulated limb emission spectra for two microwindows within the PAN $\nu_{16}$ band centred near 792 cm⁻¹ for a tangent altitude of 8 km **(a)** and the error budget for a PAN vertical profile obtained on 13 September 2017 at 16:55 UTC **(b)**. Annotation as per Fig. 2.

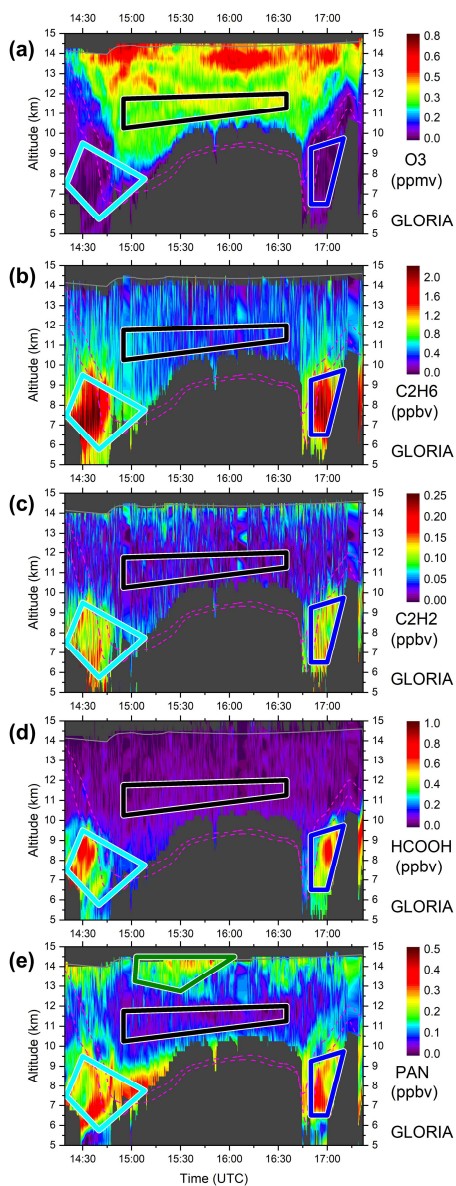

**Figure 6.** Horizontal and vertical VMR distributions of **(a)** $O_3$, **(b)** $C_2H_6$, **(c)** $(C_2H_2)$, **(d)** HCOOH, **(e)** (PAN) as seen by GLORIA above the North Atlantic region on 13 September 2017. Flight altitude is shown as grey line; dynamical tropopause (2 and 4 potential vorticity units from ECMWF) is plotted as dashed magenta lines. Cyan, blue and green (only PAN) coloured boxes mark regions with enhanced VMR levels, black boxes comprise a region with low mixing ratios. For all these boxes backward trajectories are calculated (see discussion in Sect. 3).



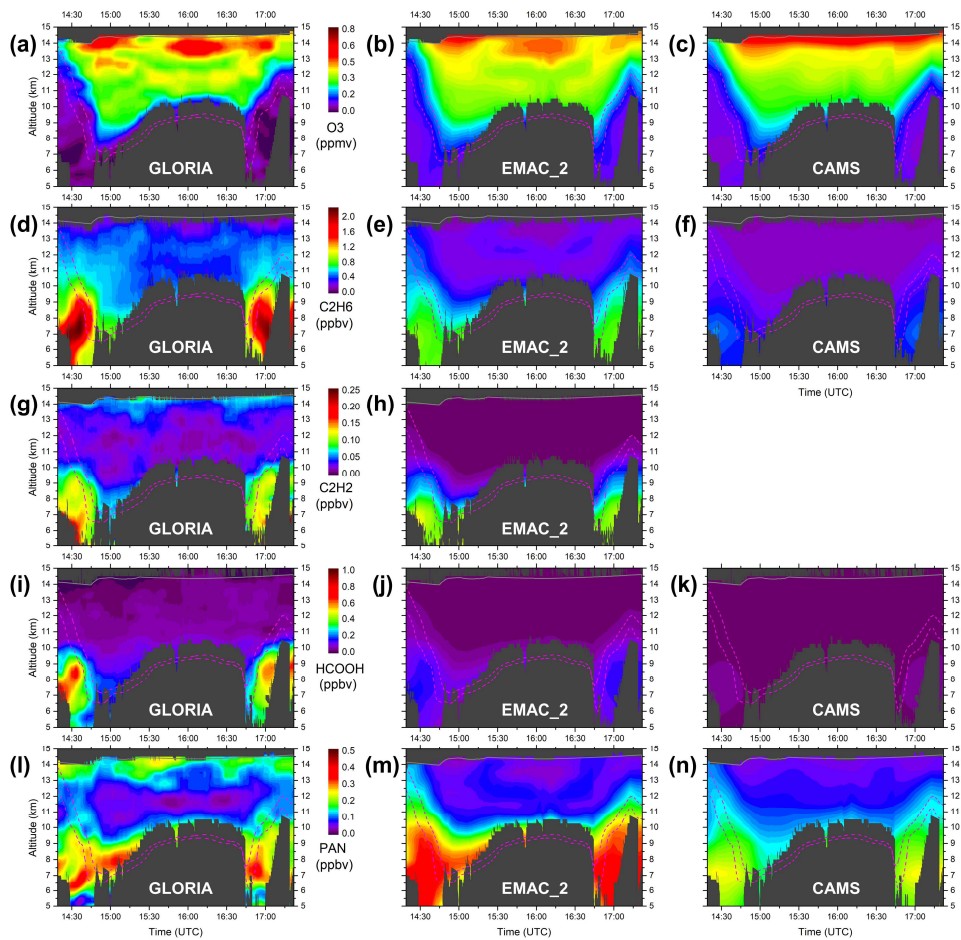

**Figure 7.** Horizontal and vertical VMR distributions of GLORIA (temporally smoothed, left column), EMAC_2 (middle column), and CAMS (right column) of **(a-c)** $O_3$, **(d-f)** $C_2H_6$, **(g-h)** $C_2H_2$, **(i-k)** HCOOH, and **(l-n)** PAN, as seen on 13 September 2017. The EMAC_2 simulation includes NMVOC emissions enhanced by a factor of 2 as recommended by Monks et al. (2018). No CAMS data for $C_2H_2$ is available. Annotation as per Fig. 6.


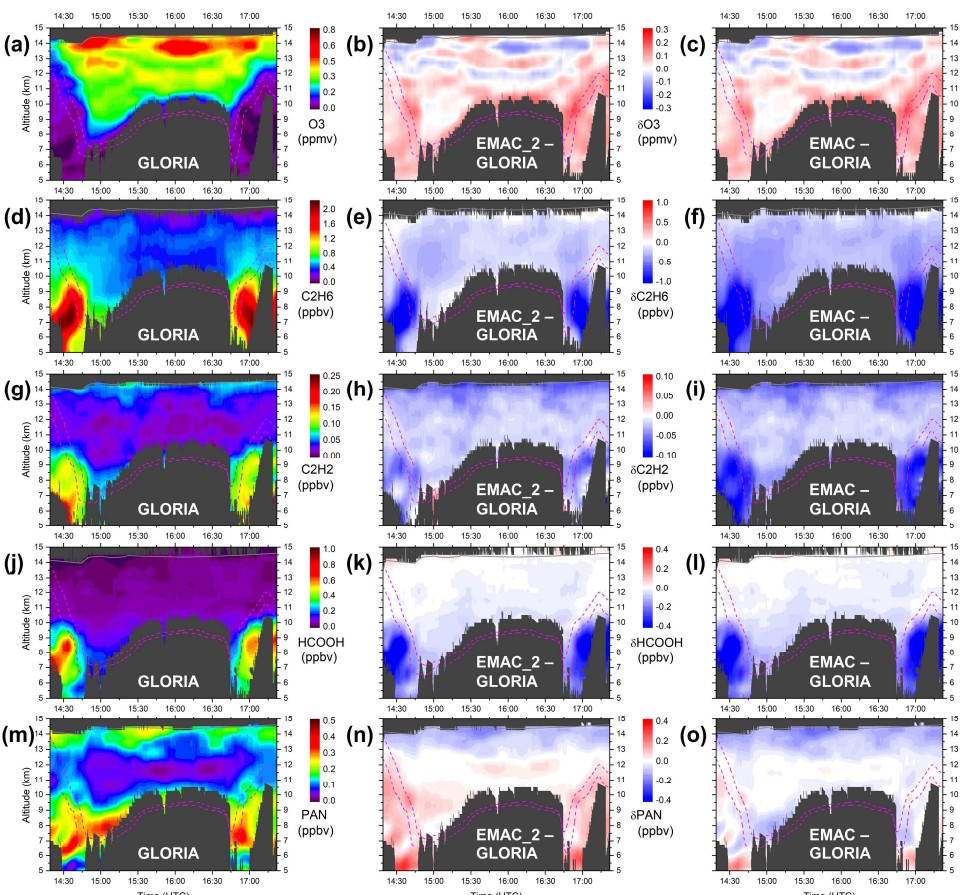

**Figure 8.** Horizontal and vertical VMR distributions of GLORIA (temporally smoothed, left column), EMAC_2 (enhanced NMVOC emissions) minus GLORIA (middle column), and EMAC (standard NMVOC emissions) minus GLORIA (right column) of **(a-c)** $O_3$, **(d-f)** $C_2H_6$, **(g-i)** $C_2H_2$, **(j-l)** HCOOH, and **(m-o)** PAN, as seen on 13 September 2017. Annotation as per Fig. 6.



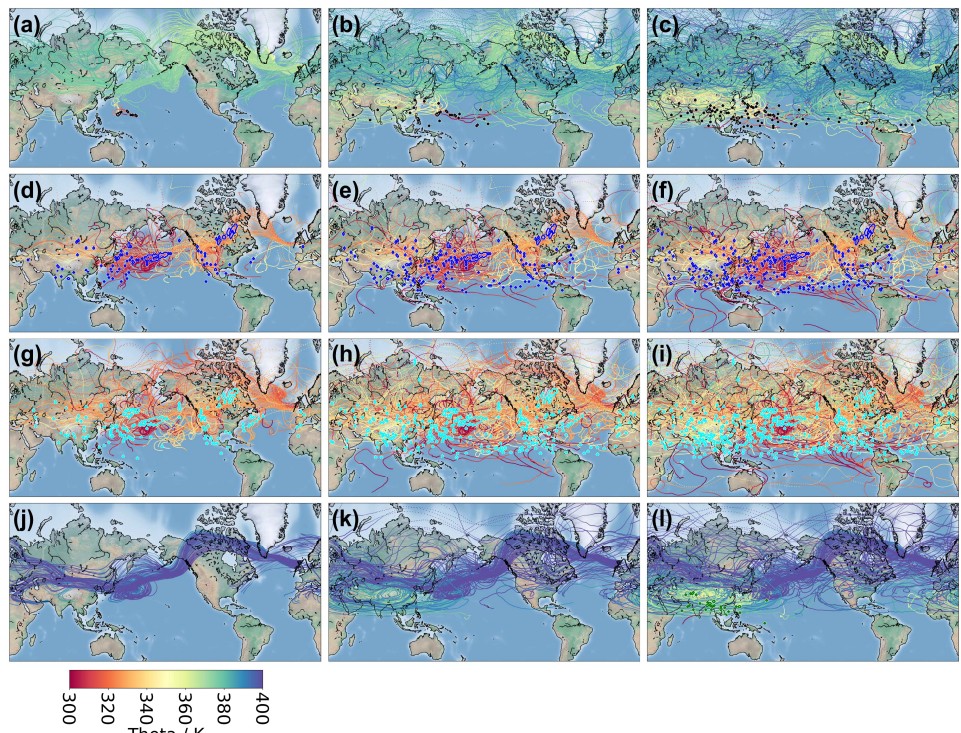

**Figure 9.** Backward trajectory calculations performed by CLaMS using ERA-interim wind data. Trajectories start at the GLORIA tangent points and are shown for 20 days (left column), 40 days (middle column), and 60 days (right column) within defined regions: black (**a-c**), blue (**d-f**), cyan (**g-i**), and green (**j-l**) as displayed in Fig. 6. Trajectory colours denote the potential temperature (which is also a measure of altitude) along the trajectory as indicated in the colour bar. Coloured encircled areas mark regions where the backward trajectory penetrates the upper edge of the planetary boundary layer (800 hPa). For better clarity, only every tenth trajectory is displayed. Hence, encircled areas may occur where no trajectory is drawn.





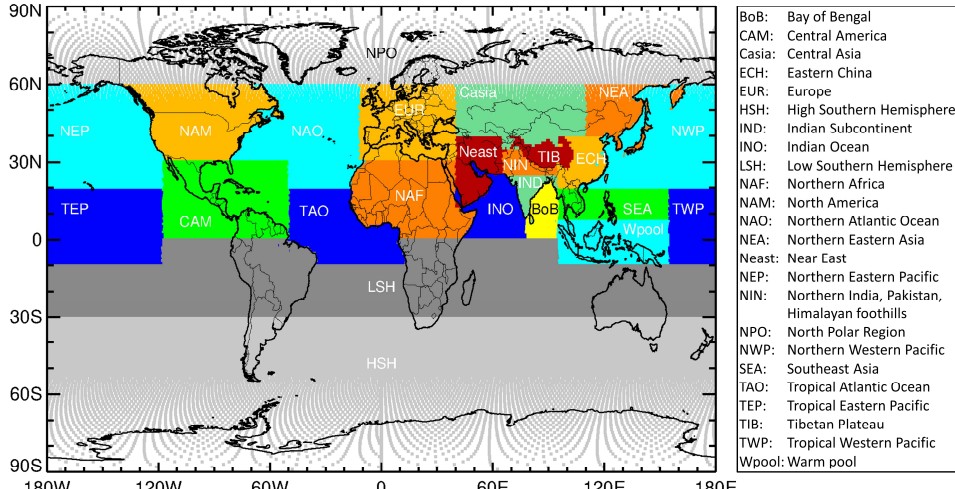

**Figure 10.** Geographical locations of the artificial tracers of air mass origin used in the CLaMS model for the HALO WISE campaign 2017. In some regions, the artificial tracers are defined to separate between continental and maritime areas as well as by different geopotential heights (e.g. Tibetan Plateau). The geographical locations of the artificial tracers of air mass origin used in previous CLaMS simulations can be found in Vogel et al. (2019).



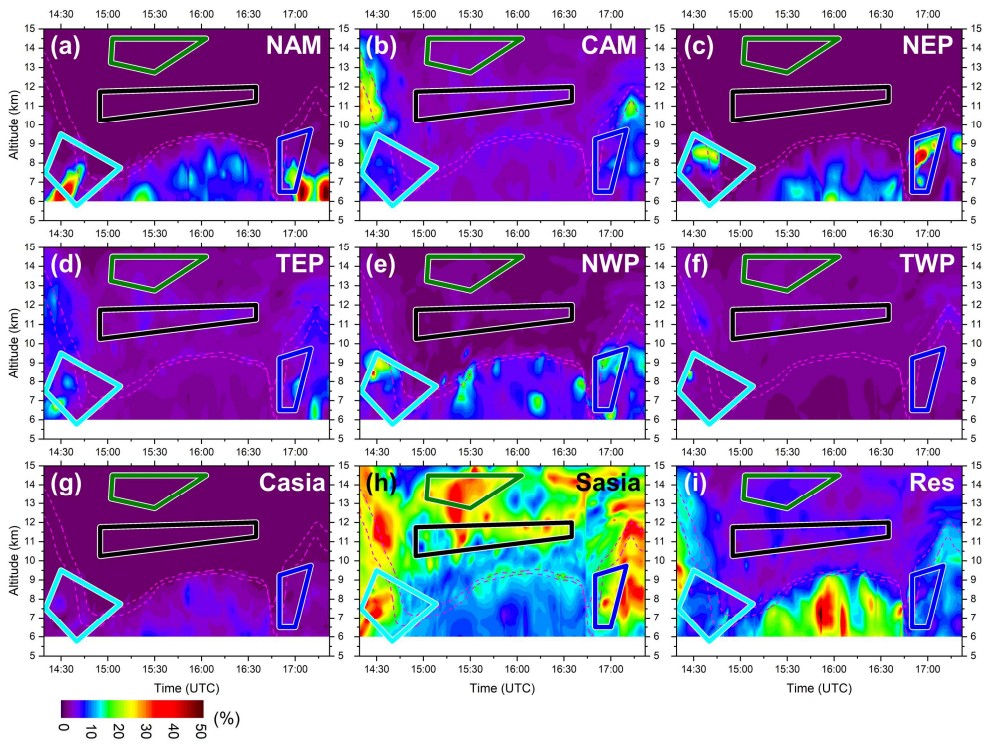


**Figure 11.** Artificial tracers of air mass origin calculations performed with CLaMS for the GLORIA observation grid showing the horizontal and vertical distribution of fraction of air originating from the boundary layer of different geographical regions as defined in Fig. 10. Results are shown for zones from the North and Central American and Pacific region (**a-f**) and zones for Central Asia (**g**) and South Asia

(Sasia) which comprises the subregions INO, IND, NIN, TIB, ECH, BoB, SEA, and Wpool (**g**). The residual part (Res) displayed in (**i**) includes all global regions except the zones shown in (**a-h**) and is of little importance for the GLORIA observations. Annotations as per Fig. 6.