# Peer review of "Pollution trace gases $C_2H_6$ , $C_2H_2$ , HCOOH, and PAN in the North Atlantic UTLS: observations and simulations"

_Atmospheric Chemistry and Physics, 2020_

## Referee Comment (RC1) · Anonymous Referee #1 · 1 Feb 2021

General comments:

This manuscript presents observations of multiple trace gases from the GLORIA instrument over the North Atlantic. It describes the GLORIA instrument and data analysis method, and then uses model simulations to interpret the observations and relate them to the Asian Monsoon and to identify underestimates in the emission inventory. This is an interesting and useful multi-species dataset and the modeling provides a robust tool for analysis. However, the combination of retrieval description and model analysis left the main focus of the manuscript a bit unclear to me. If the primary objective is to present the GLORIA retrieval and data analysis, I would like to see this section

expanded and more validation included. If the goal is instead to interpret the observations, the discussion of the driving scientific questions and what new insights are found should be clarified/strengthened.

Specific comments:

Line 60: This lifetime is still long enough to allow long-range transport

Line 132: Is radiative transport the same as radiative transfer?

Section 2.2: Is this the first description of this method, or are there other papers that could be referenced for the GLORIA retrievals and validation?

Line 216: All biomass burning or just "anthropogenic" biomass burning?

Line 275: specify the instrument/satellite

Conclusions: Last paragraph: The importance of emission inventories is well-known. Please make the conclusions more specific and emphasize what is new from this study.

Figure 6: Is this figure the same as the first column of figure 7?

Fig. 9 caption: Does "Coloured encircled areas" mean the cyan dots/circles?

Editorial comments:

Lines 46-50: please reword this sentence for better clarity

---

## Referee Comment (RC2) · Anonymous Referee #2 · 8 Feb 2021

Review of Pollution trace gases C2H6 , C2H2 , HCOOH, and PAN in the North Atlantic UTLS: observations and simulations

by Wetzel and colleagues

—- General comments

This is a nice paper that showcases measurements of upper troposphere and lower stratosphere composition by the relatively new airborne remote sensing "GLORIA" instrument. The paper does a nice job of describing these observations and using state-of-the-art models and analysis techniques to explore the likely origins of and explanations for the observed abundances.

[Figure]

Overall, I think this paper is a nice addition to the field, and provides a nice example of the GLORIA capabilities. I fully expect that it will be ready for publication once my comments below (and those from any other reviewers) have been attended to.

My only general comment is that it would have been good to include some discussion of how these observations compare to past observations of these species. Currently this is limited to comparisons to limited-resolution spaceborne remote sounding observations (e.g., the many citations to the Rinsland, Glatthor and Wiegele papers in the manuscript). However, there are a wealth of airborne in-situ observations of many these species in past campaigns (for example the NASA ATOM campaign, among many others from the US, Europe and Asia). Given that GLORIA is a relatively new (but very welcome) addition to the worldwide portfolio of airborne instruments measuring atmospheric composition, and that it is one of the few employing remote-sounding (particularly for such a wide range of species), and further, given the general skepticism some in the community have toward remote sounding observations, some additional statements as to how the GLORIA findings compare to available in situ observations of the same species at similar altitudes/latitudes/seasons, etc. would help cement the value of the GLORIA dataset in the community mind set.

—- Specific comments

Sentence spanning lines 24/25: Reword to "Elevated quantities of PAN were measured even in the lowermost stratosphere (locally up to 14 km), likely reflecting the fact that this molecule has the longest lifetime of the four species discussed herein."

Sentence spanning lines 43-46: Better to split into two sentences along the lines of "... conditions. In particular, rapid vertical transport by deep convection followed by strong horizontal transport associated with the upper troposphere subtropical jet stream ([citations]) is a particularly efficient means by which surface pollutants can be transported long distances.

Line 47: Think that the "that" would be better as ", which" in this case.

[Figure]

Line 56: "such that C2H6 may be" -> "enabling it to be"

Line 64: "are important <contributors to the tropospheric abundances of this molecule>" or something similar.

Line 66: "like" -> "such as the"

Line 105: "using" -> "observing" (to avoid having "using" twice in quick succession)

Line 154: I think something like "Test retrievals were used to identify microwindows that combine limited overlap of spectral signatures of disturbing gases with a high sensitivity to changes in the abundance of target gases." would be better wording.

Line 252: comma needed after "that"

Lines 257-287: As discussed above, it would be good to compare a small number of the wealth finding from airborne in-situ observations of these species.

Line 262: "stronger enhanced" -> "strong enhancements of"

Line 263: "what" -> ", which"

Line 265: "picture" -> "behavior"

Line 291: "with respect to" -> "given the", also "in" -> "of"

Line 292: "Concerning" -> "For"

Line 293: "principally" -> "generally"

Line 302: "The comparison of" -> "Comparisons for"

Line 311: "For C2H2 we note that EMAC predicts elevated concentrations in much the same region where GLORIA reports enhancements (see ...)"

Line 311-317: Again, this would be a good place to mention in situ comparisons.

Line 430-434: The sentence "However, the real...". I'm afraid I don't understand what

this sentence is trying to say. Does CLAMS not have emissions for these specific species on some kind of fine spatial resolution (EDGAR, MEGAN, etc.?) Please clarify what is meant by " the real regions".

Line 461: Remove "(primarily CAMS)" and add ", particularly for CAMS" at the end of the sentence.

Line 525: Some weird cut and paste typo in citation

Line 679: Extra space between "O" and "3" in citation

Line 746: "n/a-n/a" in citation.

Figures 2-5 are nicely put together.

Figure 6 (and 7): The grey line is hard to see, make it thicker. The dashed magenta line is very hard to see. I suggest you make it white and thicker (and possibly not dashed?). "...mark regions with enhanced VMR levels" - not for O3, perhaps clarify "primary pollutant VMRs" or something like that?

Figure 9: This is very hard to see given the colored continents/oceans. As pretty as they are I'd suggest a grey-scale version of the background image, or ideally just white oceans and very pale grey continents (single color, no mountains or things like that).

[Figure]

---

## Referee Comment (RC3) · Anonymous Referee #3 · 9 Feb 2021

This paper presents observations from a new airborne remote sensing instrument, GLORIA. Results from several different models are compared to the observations, with the aim to demonstrate the reasonableness of the observations, as well as to identify the sources of the features of the observed distributions. The model-observation comparisons are also presented as an evaluation of the models. I feel the paper needs some modification prior to publication, as discussed further below.

The presentation of these new observations, along with the description of the measurement technique, is worthwhile. However, the goals of the paper should be made clearer. The uniqueness of the observations could be more strongly emphasized. It

[Figure]

would be helpful to have some sort of validation of the observations through comparison to aircraft data from established measurement techniques.

The introduction seems rather awkward, with the discussion of the measured compounds seeming rather disjointed. Perhaps more discussion of the measurement technique and its uniqueness would be more appealing to readers, and then an explanation of why these species are discussed - driven by the capability to measure them. The explanation of their role in atmospheric chemistry could be saved for the analysis discussion. At l. 41, PAN is a 'secondary pollutant', not 'secondary order'. At l. 51, in what sense is ethane 'most important'?

The purpose of the model results in the paper should be made clearer. Are they being used to provide validation of the observations? It would be more appropriate to just use the model to explain the distributions and identify the sources of high mixing ratios.

Using 60-day back trajectories seems rather a stretch. I would not think they are reliable that far back. The forward CLAMS simulations of various regional tracers seem more reliable, so the back trajectories seem unnecessary.

The conclusions seem to discuss more the model evaluation aspects of the observation-model comparisons, which I do not find fully justified by the presentation of the results.
* * *

---

## Author Comment (AC1) · 15 Mar 2021

**Response to Referee #1:**

First of all we thank the referee for the effort to carefully reading the manuscript and for all comments.

**General comments:**

*This manuscript presents observations of multiple trace gases from the GLORIA instrument over the North Atlantic. It describes the GLORIA instrument and data analysis*
*method, and then uses model simulations to interpret the observations and relate them to the Asian Monsoon and to identify underestimates in the emission inventory. This is an interesting and useful multi-species dataset and the modeling provides a robust tool for analysis. However, the combination of retrieval description and model analysis left the main focus of the manuscript a bit unclear to me. If the primary objective is to present the GLORIA retrieval and data analysis, I would like to see this section expanded and more validation included. If the goal is instead to interpret the observations, the discussion of the driving scientific questions and what new insights are found should be clarified/strengthened.*

The objective is both, to present the GLORIA data retrieval of pollutant species together with an interpretation of the data with the help of model simulations. A detailed description of the general retrieval process is given in a previous paper by Johansson et al. (2018) which is cited in the manuscript. We now mention this more clearly in Section 2.2. We also included some more sentences in the conclusion part to better strengthen the messages of this study.

**Specific comments:**

*Line 60: This lifetime is still long enough to allow long-range transport.*

We included a corresponding clause to make this clear.

*Line 132: Is radiative transport the same as radiative transfer?*

Yes, we replaced "transport" by "transfer".

*Section 2.2: Is this the first description of this method, or are there other papers that could be referenced for the GLORIA retrievals and validation?*

A detailed description of the general GLORIA retrieval process is given in the paper by Johansson et al. (2018) that is already cited later in the text. We included a clarifying sentence on this issue at the end of the first paragraph in Section 2.2. This paper also contains the validation of major species observed by GLORIA (e.g. O3, HNO3, and

ClONO2). For species discussed by this paper, we lack of in-situ data to compare our GLORIA measurements with.

*Line 216: All biomass burning or just "anthropogenic" biomass burning?*

We deleted "anthropogenic" in the corresponding sentence.

*Line 275: Specify the instrument/satellite.*

The instruments are the Atmospheric Chemistry Experiment (ACE) Fourier transform spectrometer on SCISAT-1 and the Michelson Interferometer for Passive Atmospheric Sounding (MIPAS) aboard the Envisat satellite. We added a corresponding clause in the text.

*Conclusions: Last paragraph: The importance of emission inventories is well-known. Please make the conclusions more specific and emphasize what is new from this study.*

We included some more information in the conclusions to better emphasize what is new and important in our study. For instance, we emphasized that enhanced amounts of pollutant species were measured in the upper troposphere with high temporal and spatial resolution. Furthermore, it is important to state that these enhancements were detected far away from the emission sources of these species.

*Figure 6: Is this figure the same as the first column of Figure 7?*

It is not exactly the same. Figure 6 contains the unsmoothed GLORIA data while the first column of Figure 7 shows the temporally smoothed GLORIA data (as noted now more clearly in the figure caption).

*Fig. 9 caption: Does "Coloured encircled areas" mean the cyan dots/circles?*

Not only cyan, but also black, blue and green. We added this in the figure caption text for better clarity.

**Editorial comments:**

*Lines 46-50: please reword this sentence for better clarity.*

We split the sentence into two parts for better clarity.

**References**

Johansson, S., Woiwode, W., Höpfner, M., Friedl-Vallon, F., Kleinert, A., Kretschmer, E., Latzko, T., Orphal, J., Preusse, P., Ungermann, J., Santee, M. L., Jurkat-Witschas, T., Marsing, A., Voigt, C., Giez, A., Krämer, M., Rolf, C., Zahn, A., Engel, A., Sinnhuber, B.-M., and Oelhaf, H.: Airborne limb-imaging measurements of temperature, HNO3, O3, ClONO2, H2O and CFC-12 during the Arctic winter 2015/2016: characterization, in situ validation and comparison to Aura/MLS, Atmos. Meas. Tech., 11, 4737–4756, https://doi.org/10.5194/amt-11-4737-2018, 2018.

---

## Author Comment (AC2) · 15 Mar 2021

**Response to Referee #2:**

First of all we thank the referee for the effort to carefully reading the manuscript and for all comments.

**General comments:**

*My only general comment is that it would have been good to include some discussion of how these observations compare to past observations of these species. Currently*

*this is limited to comparisons to limited-resolution spaceborne remote sounding observations (e.g., the many citations to the Rinsland, Glatthor and Wiegele papers in the manuscript). However, there are a wealth of airborne in-situ observations of many these species in past campaigns (for example the NASA ATOM campaign, among many others from the US, Europe and Asia). Given that GLORIA is a relatively new (but very welcome) addition to the worldwide portfolio of airborne instruments measuring atmospheric composition, and that it is one of the few employing remote-sounding (particularly for such a wide range of species), and further, given the general skepticism some in the community have toward remote sounding observations, some additional statements as to how the GLORIA findings compare to available in situ observations of the same species at similar altitudes/latitudes/seasons, etc. would help cement the value of the GLORIA dataset in the community mind set.*

We included some discussion on the comparison to in-situ aircraft measurements in Section 3.1. and added corresponding references. We do not have a direct comparison available (in the sense of a validation) of the pollutant species retrieved by GORIA with in-situ observations. However, retrieved GLORIA amounts of these trace gases are within the spread of values measured by in-situ instruments.

**Specific comments:**

*Sentence spanning lines 24/25: Reword to "Elevated quantities of PAN were measured even in the lowermost stratosphere (locally up to 14 km), likely reflecting the fact that this molecule has the longest lifetime of the four species discussed herein."*

We changed the text according to the reviewer's suggestion.

*Sentence spanning lines 43-46: Better to split into two sentences along the lines of "... conditions. In particular, rapid vertical transport by deep convection followed by strong horizontal transport associated with the upper troposphere subtropical jet stream ([citations]) is a particularly efficient means by which surface pollutants can be transported long distances.*

We changed the text accordingly.

*Line 47: Think that the "that" would be better as ", which" in this case.*

We split the sentence into two parts for better clarity.

*Line 56: "such that C2H6 may be" -> "enabling it to be"*

We changed this part accordingly.

*Line 64: "are important contributors to the tropospheric abundances of this molecule" or something similar.*

We changed the sentence accordingly.

*Line 66: "like" -> "such as the"*

Changed.

*Line 105: "using" -> "observing" (to avoid having "using" twice in quick succession)*

Changed.

*Line 154: I think something like "Test retrievals were used to identify microwindows that combine limited overlap of spectral signatures of disturbing gases with a high sensitivity to changes in the abundance of target gases." would be better wording.*

We modified the text accordingly.

*Line 252: comma needed after "that"*

Changed.

*Lines 257-287: As discussed above, it would be good to compare a small number of the wealth finding from airborne in-situ observations of these species.*

We included text and citations of airborne in-situ observations for all pollutant species.

*Line 262: "stronger enhanced" -> "strong enhancements of"*

Changed.

*Line 263: "what" -> ", which"*

Changed.

*Line 265: "picture" -> "behavior"*

Changed.

*Line 291: "with respect to" -> "given the", also "in" -> "of"*

Changed.

*Line 292: "Concerning" -> "For"*

Changed.

*Line 293: "principally" -> "generally"*

Changed.

*Line 302: "The comparison of" -> "Comparisons for"*

Changed.

*Line 311: "For C2H2 we note that EMAC predicts elevated concentrations in much the same region where GLORIA reports enhancements (see ...)"*

Changed.

*Line 311-317: Again, this would be a good place to mention in situ comparisons.*

We now mention the in-situ airborne measurements in Section 3.1.

*Line 430-434: The sentence "However, the real...". I'm afraid I don't understand what this sentence is trying to say. Does CLAMS not have emissions for these specific species on some kind of fine spatial resolution (EDGAR, MEGAN, etc.?) Please clarify what is meant by "the real regions".*

"Real" means where emissions really occur. The original text was misleading, so we rewrote this clause for better understanding.

*Line 461: Remove "(primarily CAMS)" and add ", particularly for CAMS" at the end of the sentence.*

Changed.

*Line 525: Some weird cut and paste typo in citation.*

Okay, corrected in bibliography.

*Line 679: Extra space between "O" and "3" in citation.*

Okay, corrected in bibliography.

*Line 746: "n/a-n/a" in citation.*

Okay, corrected in bibliography.

*Figures 2-5 are nicely put together.*

Thanks.

*Figure 6 (and 7): The grey line is hard to see, make it thicker. The dashed magenta line is very hard to see. I suggest you make it white and thicker (and possibly not dashed?). "...mark regions with enhanced VMR levels" - not for O3, perhaps clarify "primary pollutant VMRs" or something like that?*

We made the grey lines and the magenta lines thicker so they are clearly visible now (white instead of magenta is not a good option for these lines because they are also included in the following Figure 8 with difference plots containing large white areas). We changed Figures 6 to 8 and the Figure caption 6 according to the reviewer's suggestion.

*Figure 9: This is very hard to see given the colored continents/oceans. As pretty as they are I'd suggest a grey-scale version of the background image, or ideally just white*

*oceans and very pale grey continents (single color, no mountains or things like that).*

It is a difficult task to find a common optimal background for all individual pictures displayed in Figure 9. We already tested a lot of background and color scale combinations before submitting this manuscript. We again tested many combinations and found that white oceans is not the best solution because yellowish colors are hardly visible. In the end we omitted the surface relief and took for continents a light grey and for oceans a light blue color and changed Figure 9 accordingly.

---

## Author Comment (AC3) · 15 Mar 2021

**Response to Referee #3:**

First of all we thank the referee for the effort to carefully reading the manuscript and for all comments. Citations mentioned below are included in the manuscript.

**General comments:**

*The presentation of these new observations, along with the description of the measurement technique, is worthwhile. However, the goals of the paper should be made*

*clearer. The uniqueness of the observations could be more strongly emphasized. It would be helpful to have some sort of validation of the observations through comparison to aircraft data from established measurement techniques.*

We included additional motivation in the introduction and in the conclusions to better emphasize the goals of this paper. Since no co-incident in-situ observations enabling a direct comparison (in the sense of a validation) are available, we have checked in detail published airborne datasets concerning the pollution trace gases derived from GLORIA. We can conclude that retrieved GLORIA amounts of these trace gases are within the spread of values measured by in-situ instruments. Related citations are now provided in Section 3.1 individually for each species.

*The introduction seems rather awkward, with the discussion of the measured compounds seeming rather disjointed. Perhaps more discussion of the measurement technique and its uniqueness would be more appealing to readers, and then an explanation of why these species are discussed - driven by the capability to measure them. The explanation of their role in atmospheric chemistry could be saved for the analysis discussion. At l. 41, PAN is a 'secondary pollutant', not 'secondary order'. At l. 51, in what sense is ethane 'most important'?*

We include and modified sentences in the introduction to make the text and the goal of these measurements clearer. A comprehensive description of the technical issues of GLORIA is given in the papers by Friedl-Vallon et al. (2014) and Riese et al. (2014) and references therein. This is written now more clearly in the text. We find that the description of the pollutant species is better suited for the introduction than for the data analysis section (of course, this approach may be a matter of taste). In line 41 we omitted "order". Line 51: It is the most important non-methane hydrocarbon constituent of natural gas. We modified this clause accordingly.

*The purpose of the model results in the paper should be made clearer. Are they being used to provide validation of the observations? It would be more appropriate to just*

*use the model to explain the distributions and identify the sources of high mixing ratios.*

Of course, the use of the models is not intended to validate the measurements. It should be understood as a kind of intercomparison. One goal was to see if the models EMAC and CAMS are capable to reproduce the locations of the enhanced amounts of pollutants. Another goal was to quantify the differences between measured and simulated data, especially in the case of EMAC where we performed different emission scenarios. Concerning the origin of the detected enhancements, we used backward trajectories and artificial tracers of air mass origin calculations. We tried to make this issue clearer in the revised text.

*Using 60-day back trajectories seems rather a stretch. I would not think they are reliable that far back. The forward CLAMS simulations of various regional tracers seem more reliable, so the back trajectories seem unnecessary.*

We agree with the referee, that in general, trajectory calculations have limitations due to trajectory dispersion depending on the trajectory length. Therefore the uncertainty of a single trajectory is increasing with the trajectory length, however the variability of a cluster of trajectories starting in the same region, reflects the impact of mixing processes. Therefore, in this study a plenty of back-trajectories are started in the marked regions with enhanced VMR levels. Frequently employed trajectory lengths to study transport processes in particular in the Asian monsoon region range from a couple of weeks to a few months depending on the transport times from Earth's surface to atmospheric altitudes (e.g., Chen et al., 2012; Bergman et al., 2013; Garny and Randel, 2016; Müller et al., 2016; Li et al., 2018; Vogel et al., 2019; Legras and Bucci, 2020; Hanumanthu et al., 2020). In particular the CLaMS backward calculations to analyse the regions with enhanced PAN between 13 and 14.5 km (about 400 K) demonstrate that the transport times from the planetary boundary layer in Asia to the extratropical UTLS over the Atlantic are between 40 and 60 days. Therefore, trajectories up to a lengths of 60 days are necessary in our study to infer the possible source regions of PAN. Further, the endpoints of the trajectories in the planetary boundary layer show

a good overall agreement to the results of forward CLaMS simulations for various regional tracers demonstrating that trajectories of a length up to 60 days are suitable for our study.

Moreover, back trajectory calculations have an added valued to the 3-dimensional forward calculations with CLaMS because they demonstrate the detailed transport pathways from the boundary source to the locations of the measurement and its transport times. The trajectory calculations show that the air parcels with enhanced PAN are uplifted by diabatic heating in the upward spiralling range (e.g., Vogel et al., 2019) of the Asian monsoon anticyclone up to about 400 K within about 40 days and subsequent transport (within about 20 days) occurred along the subtropical jet to the extratropical UTLS over the Atlantic. This detailed transport pathway and its transport time cannot be inferred from CLaMS 3-dimensional forward calculations, therefore the back-trajectory analysis is an added value to our study.

*The conclusions seem to discuss more the model evaluation aspects of the observation-model comparisons, which I do not find fully justified by the presentation of the results.*

The conclusion consists of two main parts. First the findings of the measurements and second, the findings connected with the comparison to the models EMAC and CAMS. We included some text to make this issue more clearly.

**References**

Bergman, J. W., Fierli, F., Jensen, E. J., Honomichl, S., and Pan, L. L.: Boundary layer sources for the Asian anticyclone: Regional contributions to a vertical conduit, J. Geophys. Res. Atmos., 118, 2560–2575, https://doi.org/10.1002/jgrd.50142, 2013.

Chen, B., Xu, X. D., Yang, S., and Zhao, T. L.: Climatological perspectives of air transport from atmospheric boundary layer to tropopause layer over Asian monsoon regions during boreal summer inferred from Lagrangian approach, Atmos. Chem. Phys., 12,
5827–5839, https://doi.org/10.5194/acp-12-5827-2012, 2012.

Friedl-Vallon, F., Gulde, T., Hase, F., Kleinert, A., Kulessa, T., Maucher, G., Neubert, T., Olschewski, F., Piesch, C., Preusse, P., Rongen, H., Sartorius, C., Schneider, H., Schönfeld, A., Tan, V., Bayer, N., Blank, J., Dapp, R., Ebersoldt, A., Fischer, H., Graf, F., Guggenmoser, T., Höpfner, M., Kaufmann, M., Kretschmer, E., Latzko, T., Nordmeyer, H., Oelhaf, H., Orphal, J., Riese, M., Schardt, G., Schillings, J., Sha, M. K., Suminska-Ebersoldt, O., and Ungermann, J.: Instrument concept of the imaging Fourier transform spectrometer GLORIA, Atmos. Meas. Tech., 7, 3565–3577, https://doi.org/10.5194/amt-7-3565-2014, 2014.

Garny, H. and Randel, W. J.: Transport pathways from the Asian monsoon anticyclone to the stratosphere, Atmos. Chem. Phys., 16, 2703–2718, https://doi.org/10.5194/acp-16-2703-2016, 2016.

Hanumanthu, S., Vogel, B., Müller, R., Brunamonti, S., Fadnavis, S., Li, D., Ölsner, P., Naja, M., Singh, B. B., Kumar, K. R., Sonbawne, S., Jauhiainen, H., Vömel, H., Luo, B., Jorge, T., Wienhold, F. G., Dirkson, R., and Peter, T.: Strong day-to-day variability of the Asian Tropopause Aerosol Layer (ATAL) in August 2016 at the Himalayan foothills, Atmos. Chem. Phys., 20, 14273–14302, https://doi.org/10.5194/acp-20-14273-2020, 2020.

Legras, B. and Bucci, S.: Confinement of air in the Asian monsoon anticyclone and pathways of convective air to the stratosphere during the summer season, Atmos. Chem. Phys., 20, 11045–11064, https://doi.org/10.5194/acp-20-11045-2020, 2020.

Li, D., Vogel, B., Müller, R., Bian, J., Günther, G., Li, Q., Zhang, J., Bai, Z., Vömel, H., and Riese, M.: High tropospheric ozone in Lhasa within the Asian summer monsoon anticyclone in 2013: influence of convective transport and stratospheric intrusions, Atmos. Chem. Phys., 18, 17979–17994, https://doi.org/10.5194/acp-18-17979-2018, 2018.
Müller, S., Hoor, P., Bozem, H., Gute, E., Vogel, B., Zahn, A., Bönisch, H., Keber, T., Krämer, M., Rolf, C., Riese, M., Schlager, H., and Engel, A.: Impact of the Asian monsoon on the extratropical lower stratosphere: trace gas observations during TACTS over Europe 2012, Atmos. Chem. Phys., 16, 10573–10589, https://doi.org/10.5194/acp-16-10573-2016, 2016.

Riese, M., Oelhaf, H., Preusse, P., Blank, J., Ern, M., Friedl-Vallon, F., Fischer, H., Guggenmoser, T., Höpfner, M., Hoor, P., Kaufmann, M., Orphal, J., Plöger, F., Spang, R., Suminska-Ebersoldt, O., Ungermann, J., Vogel, B., and Woiwode, W.: Gimballed Limb Observer for Radiance Imaging of the Atmosphere (GLORIA) scientific objectives, Atmos. Meas. Tech., 7, 1915–1928, https://doi.org/10.5194/amt-7-1915-2014, 2014.

Vogel, B., Müller, R., Günther, G., Spang, R., Hanumanthu, S., Li, D., Riese, M., and Stiller, G. P.: Lagrangian simulations of the transport of young air masses to the top of the Asian monsoon anticyclone and into the tropical pipe, Atmos. Chem. Phys., 19, 6007–6034, https://doi.org/10.5194/acp-19-6007-2019, 2019.

---

## Author Response (AR2)

**Author's response file**

In the following, we list the answers to the report from Referee #1 together with the complete revised manuscript. Changes in the text of the manuscript with respect to the previous version are marked with green colour (changes in red colour refer to the original manuscript as published in ACPD)

**Response to report from Referee #1 (minor revisions):**

*The authors have made the goals of the study clearer in the revised manuscript by adding text in the introduction and conclusions. I feel the paper could be further improved by a few changes to the presentation to emphasize what is unique in this paper. For example, when citing Johansson et al. (2018), it would be helpful to state explicitly what is new/different here, i.e. the focus on different chemical species.*

The mentioned paper by Johansson et al. (2018) contains a detailed description of the general retrieval process and validation of trace gases which are somewhat easier to analyse compared to the pollution trace species discussed within the present study. We included further text at line 163 et seq. to mention that these pollution gases are more challenging to retrieve compared to those described in Johansson et al. (2018). Hence, it should be clear now that the subsequent sections describing the retrieval process of the pollution species is the "new/different" information given in this paper.

*The additional references to aircraft observations are a nice addition, but perhaps organizing this information into a table showing the GLORIA observations compared to the range of values in the literature, or at least describing the range in the text, for each species would make the comparison easier and emphasize the value of these new observations. With these additions, I think the paper would make a nice addition to the literature.*

We included more precise information on the specific values measured by other instruments in the corresponding text of the pollutant trace gases in Section 3.1.

[revised manuscript text omitted]